# Avoiding organelle mutational meltdown across eukaryotes with or without a germline bottleneck

David M. Edwards[1], Ellen C. Røyrvik[2], Joanna M. Chustecki[3], Konstantinos Giannakis[4], Robert C. Glastad[4], Arunas L. Radzvilavicius[4], Iain G. Johnston[4,5]*

1 School of Life Sciences, University of Warwick, United Kingdom, 2 Department of Clinical Science, University of Bergen, Norway, 3 School of Biosciences, University of Birmingham, United Kingdom, 4 Department of Mathematics, University of Bergen, Norway, 5 Computational Biology Unit, University of Bergen, Norway

* iain.johnston@uib.no

**Data Availability Statement:** Code and data are available at https://github.com/StochasticBiology/odna-segregation (DOI: 10.5281/zenodo.4572545).

## Abstract

Mitochondrial DNA (mtDNA) and plastid DNA (ptDNA) encode vital bioenergetic apparatus, and mutations in these organelle DNA (oDNA) molecules can be devastating. In the germline of several animals, a genetic "bottleneck" increases cell-to-cell variance in mtDNA heteroplasmy, allowing purifying selection to act to maintain low proportions of mutant mtDNA. However, most eukaryotes do not sequester a germline early in development, and even the animal bottleneck remains poorly understood. How then do eukaryotic organelles avoid Muller's ratchet—the gradual buildup of deleterious oDNA mutations? Here, we construct a comprehensive and predictive genetic model, quantitatively describing how different mechanisms segregate and decrease oDNA damage across eukaryotes. We apply this comprehensive theory to characterise the animal bottleneck with recent single-cell observations in diverse mouse models. Further, we show that gene conversion is a particularly powerful mechanism to increase beneficial cell-to-cell variance without depleting oDNA copy number, explaining the benefit of observed oDNA recombination in diverse organisms which do not sequester animal-like germlines (for example, sponges, corals, fungi, and plants). Genomic, transcriptomic, and structural datasets across eukaryotes support this mechanism for generating beneficial variance without a germline bottleneck. This framework explains puzzling oDNA differences across taxa, suggesting how Muller's ratchet is avoided in different eukaryotes.

## Introduction

Mitochondrial DNA (mtDNA) and plastid DNA (ptDNA) play vital roles in eukaryotic cells [1,2]. mtDNA encodes bioenergetic machinery in eukaryotes, including the most central aspects of the electron transport chain [3]. ptDNA encodes many of the core proteins of the complexes involved in photosynthetic electron transfer in eukaryotic photoautotrophs.

**Funding:** DME and JMC are supported by the BBSRC via the MIBTP Doctoral Training Scheme. This project has received funding from the European Research Council (ERC) under the European Union's Horizon 2020 research and innovation programme (Grant agreement No. 805046 (EvoConBiO) to IGJ). The funders had no role in study design, data collection and analysis, decision to publish, or preparation of the manuscript.

**Competing interests:** The authors have declared that no competing interests exist.

**Abbreviations:** AIC, Akaike information criterion; BAR, Bio-Analytic Resource for Plant Biology; GCOS, Affymetrix GeneChip Operating Software; mtDNA, mitochondrial DNA; oDNA, organelle DNA; ptDNA, plastid DNA; SAM, shoot apical meristem; TGT, target intensity.

Because of this energetic and metabolic centrality, mutations in these organelle DNA (oDNA) molecules can be devastating, and avoiding the inheritance of mutant oDNA is an evolutionary and medical priority [4–8]. oDNA in eukaryotic cells is highly polyploid: Hundreds or thousands of oDNA molecules typically exist in a single cell, some of which may harbour mutations. The resulting mixture of oDNA types in a cell is referred to as heteroplasmic; we will refer to the proportion of mutant oDNA molecules in the cell as the heteroplasmy level. For pathological mtDNA mutations, for example, disease phenotypes appear when this heteroplasmy level exceeds a certain threshold [9].

Due to its role in human disease, mtDNA in animals is perhaps the most studied oDNA system. Here, shifts in mean heteroplasmy level can be achieved in the germline [10,11], with recent work linking this genetic shift to a physical mechanism involving the fragmentation of mitochondria [12]. But these shifts are likely unable to completely remove mutant mtDNA from a population, which may be a reason behind the ongoing presence of human mtDNA diseases [6]. As an additional strategy in several organisms, a genetic "bottleneck" in germline development induces intercellular variability in heteroplasmy level [13–15], as well as accelerating what selective mechanisms may be present [11,16]. Rather than a mother with 50% heteroplasmy level producing oocytes which each have 50% heteroplasmy level, this bottleneck segregates heteroplasmy across oocytes, producing, for example, a range from 30% to 70%. Purifying selection can then act to discard those oocytes with a higher heteroplasmy level (Fig 1A). Increasing cell-to-cell variance in heteroplasmy level thus provides a means to reduce heteroplasmy level between generations, slowing Muller's ratchet—the ongoing accumulation of deleterious mutations, leading to "mutational meltdown"—over evolutionary history [4,15,17]. Where quality control does apply intracellular selective pressure against mutations, the increase in variance acts in concert with this selection, to further increase the probability of inheriting a low heteroplasmy level [12,14,16].

Variability in oDNA populations can in principle be induced by various specific cellular mechanisms, including random partitioning of oDNA at cell divisions, the stochastic replication and degradation of oDNA, potential random samplings of oDNA within a cell (for example, allowing replication only of a subset of oDNAs and allowing the rest to degrade), the physical dynamics of organelles, and gene conversion (one oDNA molecule effectively "overwriting" another). Each of these mechanisms can cause differences in oDNA populations to arise between cells. In mammals and other animals, mtDNA segregation (the genetic bottleneck) is accelerated by a physical bottleneck [13,14,16,18,19]. Here, soon after fertilisation, mtDNA copy number per cell decreases dramatically as cells divide. This reduction in copy number amplifies the effect of cellular mechanisms inducing cell-to-cell variability, accelerating genetic drift as mtDNA molecules are partitioned and turned over. These animals have a dedicated germline, sequestered early in female development, where mtDNA depletion occurs. However, most eukaryotes, including basal metazoans like corals and sponges, do not sequester a germline in the same way [20,21] (although the extent of germline sequestration in plants, for example, is far from resolved [22]). Many lack a fixed body plan and allow somatic tissue to eventually give rise to gametes. How do such organisms avoid oDNA mutational meltdown?

There is little evidence for a physical bottleneck in these species, suggesting that other mechanisms must be used to generate evolutionarily beneficial oDNA variance [4,15,23]. In parallel, striking contrasts in behaviour of oDNA and organelles exist between mammals and these other organisms [24,25]. Plant and fungal mtDNA is recombinationally active [26–28], and dramatic shifts in the balance of mtDNA types, termed substoichiometric shifting, occur in plants [25,28,29]. mtDNA recombination surveillance genes are found in corals and sponges but not other metazoans [20]. Physically, plant mitochondria largely remain highly fragmented (and highly motile) except in the aboveground germline [25,30–32], while fused mitochondria

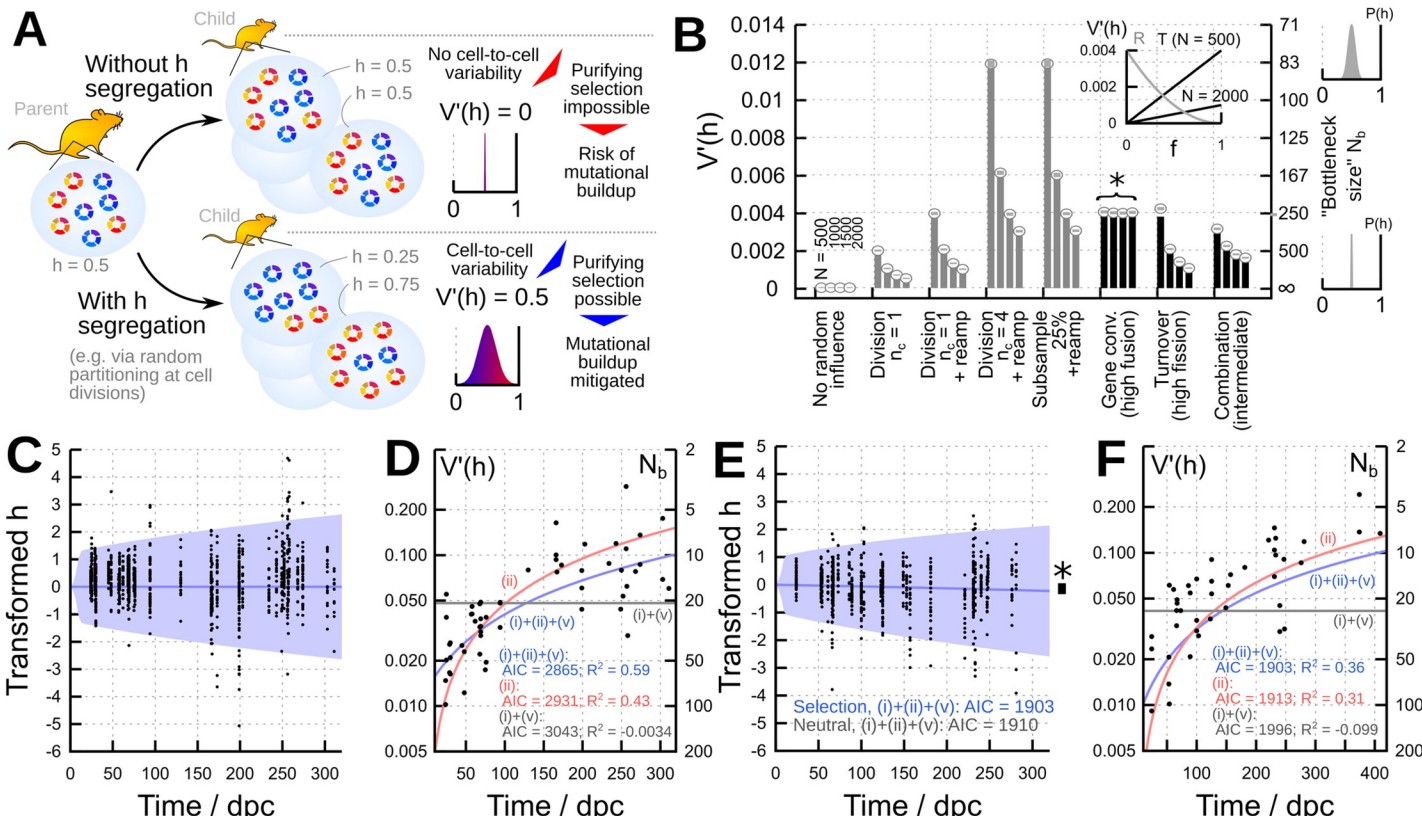

**Fig 1. Quantitative prediction of beneficial heteroplasmy variance generated by different cellular mechanisms. (A)** Beneficial mtDNA segregation avoids Muller's ratchet. Increasing cell-to-cell variance $V'(h)$ in heteroplasmy levels allows purifying selection to reduce heteroplasmy level between generations; without such variance, there is a risk of mutational meltdown over evolutionary time. **(B)** Quantitative contributions of different cellular and subcellular mechanisms to $V'(h)$ predicted from Eq 1 (columns) and observed in stochastic simulation ($10^4$ repeats, white disks, grey bars within disks give 95% confidence intervals). Grey columns are single processes; black columns are processes where variance accumulates over time (illustrated here over 1 day). $N$ is cellular oDNA population size, and illustrative parameters $\nu = 1$ **day$^{-1}$**, $\kappa = 0.002$ **day$^{-1}$**, $t = 1$ **day**. * highlights that gene conversion, unlike other mechanisms, has no dependence on population size $N$. Inset shows the relative contributions of recombination (R) and turnover (T) to $V'(h)$ (and bottleneck size $N_b$) with different organelle fragmentation fractions $f$ and population size $N$. **(C–F)** Comparison of the predictions of Eq 1 to the results of single-cell experiments tracking heteroplasmy variance in the mouse germline. Heteroplasmy is transformed to account for different initial conditions (see Methods). **(C)** Spread of heteroplasmy levels in single mouse oocytes from mothers of different ages in model HB of reference [11], and predicted mean and 95% intervals from Eq 1. **(D)** Heteroplasmy level variance $V'(h)$ and corresponding bottleneck size $N_b$ from these measurements, with predictions from different mechanisms using Eq 1 ((i) cell divisions; (ii) turnover; and (v) reamplification). **(E)** Spread of heteroplasmy levels in single mouse oocytes from mothers of different ages in model LE of reference [11], with significant germline selection (*), and predicted mean and 95% confidence intervals from the non-neutral version of our theory (see Methods). **(F)** Heteroplasmy level variance $V'(h)$ and corresponding bottleneck size $N_b$ from these measurements, with predictions from different mechanisms using the non-neutral theory ((i) cell divisions and (ii) turnover). Further model comparison plots are included in S2 Fig; for data and code, see https://github.com/StochasticBiology/odna-segregation. dpc, days post conception; mtDNA, mitochondrial DNA.

are common in fungi and animals. Plastid fusion is only observed in limited conditions [33,25], making plastids more independent compartments, although ptDNA recombination within an organelle is common [34] (and harnessed in biotechnology [35]).

The quantitative links between these diverse behaviours and the generation of beneficial oDNA variance remain to be revealed, both in animals and in more diverse eukaryotes. In mammals, progress has been made linking some mechanisms (including a physical germline bottleneck) to mtDNA heteroplasmy level variance (reviewed in reference [14]), although the contribution of different mechanisms in generating mtDNA variance remains debated [14,16,18,36,37]. Theory has explored the role of different organelle inheritance strategies, in the presence or absence of recombination, on mtDNA segregation [4], and the influence of different reproductive and inheritance strategies on mutational load has been modelled

analytically [38]. Further classical theory has shed quantitative light on the interplay of germ-line bottlenecks and selection [39], the role of oDNA variability (including that generated by gene conversion) in fixing new mutants within model cells and populations [40,41]. In plants, Lonsdale and colleagues [42] highlighted recombination as a mechanism for generating mtDNA genetic heterogeneity. Khakhlova and Bock [23] proposed that gene conversion helps plastids combat Muller's ratchet and drew a parallel between gene conversion and the physical mtDNA bottleneck in animals as means of generating beneficial organelle heterogeneity. Following these ideas, we sought here to identify which processes, in addition to or instead of a physical bottleneck, may contribute to the generation of evolutionarily beneficial heteroplasmy level variance across eukaryotes (including animals). To this end, we build a comprehensive statistical genetic framework quantifying the effects of different cellular and molecular influences on heteroplasmy level variance both in animals and more diverse eukaryotes, interrogate it to explore the interplay between development, organelle populations, and heteroplasmy level variance, and use genomic, transcriptomic, and ultrastructural data from diverse systems biology sources to support the hypotheses that emerge from this analysis.

## Results

### Diverse cellular mechanisms can achieve a "genetic bottleneck"

The result of the genetic bottleneck is a decrease in the effective cellular population size of oDNA. This increases cell-to-cell variance in heteroplasmy level (Fig 1A) and may accelerate any intracellular selection of one oDNA type over another that occurs in parallel [10–12]. Across many cells, we will observe a mean heteroplasmy level $E(h)$ and heteroplasmy level variance $V(h)$. We will consider normalised heteroplasmy level variance $V'(h) = V(h)/(E(h)(1-E(h)))$, which accounts for the dependence of heteroplasmy variance on the mean heteroplasmy level. This quantity is simply related to the "bottleneck parameter" $b = 1-V'(h)$ and is often pictured as an effective bottleneck size $N_b = 1/V'(h)$ [14]. Here, a smaller "genetic bottleneck" corresponds to more cell-to-cell variance in heteroplasmy level (Fig 1A). We sought to quantitatively investigate the hypothesis that a variety of different mechanisms, including cell divisions, oDNA turnover, and gene conversion, can increase cell-to-cell variability in oDNA heteroplasmy level and thus contribute to the genetic bottleneck.

To this end, we constructed a stochastic model describing populations of oDNA in cells and the process which can act to change these populations. We consider cells containing two oDNA types (wild-type $a$ and mutant $A$; our approach can readily be generalised to more types). This situation models, for example, oDNA types differing at one locus or a number of loci. Larger-scale structural oDNA variation also exists in many eukaryotes; our model can, with some caveats (see Discussion), be applied to this case too. We do not consider here mutational processes that generate de novo variants, although this could be readily captured within our modelling framework (see Discussion). Cellular copy numbers of these two types are $n_a$ and $n_A$, respectively, and heteroplasmy level is defined as $h = n_A/(n_a+n_A)$. Each oDNA molecule may exist in a single fragmented organelle or as part of a fused collective structure (for example, a mitochondrial network). The processes in our stochastic model correspond to the different genetic and physical mechanisms that may act on cellular oDNA populations.

These processes include the random replication and degradation of oDNA without a systematic change in copy number, which we refer to as turnover [43,44]; replication of a restricted subset of oDNAs within the cell, which we refer to as subsampling [37]; partitioning of oDNA molecules individually or in clusters at cell divisions [36,45,46]); reamplification (where a reduced oDNA population grows in size through random replication); the fusion of fragmented organelles to form a collective containing several oDNAs; the fission of a collective

into fragmented organelles containing single oDNAs; and oDNA gene conversion [23]. This final mechanism is recombination dependent and consists of one oDNA acting as a template to convert another [47–49], leading to $aA{\to}aa$ or $aA{\to}AA$ (where bias may favour one process or the other). This model thus describes the mechanisms that have been proposed to contribute to the mtDNA bottleneck in animals [14,16,18,36,37] and to oDNA variability in more diverse eukaryotes [23,42], in addition to accounting for the physical embedding of oDNAs within organelles [50].

In our model, each of these processes occurs with some rate, which may be zero (in which case the corresponding process plays no role in the model) and may depend on the current number of oDNAs in the cell (for example, allowing the cell to control oDNA replication [43,44]). The cell-to-cell variance generated by each of these processes is a mathematical function of these rates and some additional parameters describing each mechanism (see below). We derive the variance contribution of each mechanism in our model using tools from the mathematics of stochastic processes, summarised in Methods and carried out in S1 Text. Briefly, for intracellular processes, we write down the "chemical master equation" describing how each process influences a cellular oDNA population, then approximate this equation with a differential equation describing how the resultant heteroplasmy statistics change over time. For sampling processes (including replication of a subset of oDNAs and partitioning at cell divisions), we use the statistics of probability distributions corresponding to each type of sampling. After deriving these expressions, we can analyse our model to ask how much cell-to-cell variance is generated by the action of each different mechanism, allowing a quantitative investigation of our hypothesis.

The first result from this framework is, to our knowledge, the first quantitative description of how beneficial variance in heteroplasmy level is generated by each of these cellular and subcellular processes. This theory can be applied both to characterise the germline bottleneck found in some animals and to characterise other mechanisms generating oDNA variance in other eukaryotes. While our theory can account for the influence of nonzero selective differences between oDNA types (see below), we first focus on the case of no selective difference. Here, the quantitative contributions of each mechanism to evolutionarily beneficial heteroplasmy level variance take a remarkably simple form, given by Eq 1 in Box 1. Assuming oDNA copy numbers are always much greater than 1, and in the case where neither allele experiences a selective advantage, Eq 1 predicts to a good approximation:

---

**Box 1.** General formula for oDNA heteroplasmy level variance

$$V'(h) = 1/N_b \simeq$$

$$\underbrace{\sum_{\text{divisions } i} \frac{n_c}{n_{1,i}}}_{\text{(i) cell divisions}} + \underbrace{\sum_{\substack{\text{cell}\\ \text{cycles } i}} t_i \left( \overbrace{\frac{f_i(1+n_d)v_i}{n_{1,i}}}^{\text{(ii) turnover}} + \overbrace{2(1-f_i)^2 \kappa_i}^{\text{(iii) gene conversion}} \right)}_{\text{dynamics within cell cycles}} + \underbrace{\sum_{\substack{\text{sub}\\ \text{samples } i}} n_c \left( \frac{1}{n_{2,i}} - \frac{1}{n_{1,i}} \right)}_{\text{(iv) cellular subsampling}} + \underbrace{\sum_{\substack{\text{amplif}-\\ \text{ications } i}} n_c \left( \frac{1}{n_{1,i}} - \frac{1}{n_{2,i}} \right)}_{\text{(v) cellular reamplification}} \quad (1)$$

Here, $V'(h)$ is cell-to-cell heteroplasmy level variance (the evolutionarily beneficial quantity of interest), and the closely related statistic "bottleneck size" is $N_b = 1/V'(h)$.

Each summation sums the contribution to $V'(h)$ of any given number of events of five types: (i) partitioning of oDNA molecules at cell divisions; dynamics of oDNA during a

---

cell cycle including (ii) turnover and (iii) gene conversion; (iv) cellular subsampling of oDNA molecules; and (v) cellular amplification of oDNA molecules.

In each case, $n_{1,i}$ is the cellular oDNA copy number before event $i$ and $n_{2,i}$ is the copy number after event $i$. Parameters $v_i$, $\kappa_i$, and $f_i$ respectively describe the degradation rate, gene conversion rate, and proportion of fragmented organelles during cell cycle $i$, which is of length $t_i$. The cluster size parameter $n_c$ refers to the number of oDNA molecules that are sampled or partitioned as a single unit (for example, in physically connected nucleoids). If oDNAs are inherited and/or sampled individually, or in heteroplasmic clusters, $n_c = 1$. If oDNAs are inherited and/or sampled in homoplasmic clusters of size $c$, $n_c = c$. $n_d$ is the number of oDNA molecules that are destroyed in one autophagy event. If selection also acts on oDNA, the theory remains tractable but takes a longer form; see Methods and S1 Text.

Proposed mechanisms for the mammalian germline involve the use of (i), (ii), (iv), and (v) with different emphases (for example, reference [37] emphasises (iv); reference [18] emphasises (i)+(ii); and reference [36] emphasises (i) with high $n_c$). Eq 1 allows us to compare these proposed mechanisms and their associated strengths in the light of experimental data, as in the main text. Eukaryotes that do not deplete oDNA copy number to the same extent may compensate with the use of mechanism (iii) (which does not depend on copy number $n$), also explored in the main text.

i. Every cell division adds $n_c/n$ to the normalised variance, where $n$ is oDNA copy number before division, and $n_c$ is inherited cluster size (see Box 1);

ii. oDNA turnover adds $(1+n_d)vft/n$, where $v$ is oDNA turnover rate, $t$ is the length of time during which turnover occurs, $n$ is oDNA copy number, $f$ is proportion of fragmented mitochondria, and $n_d$ is the number of molecules destroyed per autophagy event (for example, the number of oDNAs contained in an organelle);

iii. oDNA gene conversion adds $2\kappa(1-f)^2 t$, where $\kappa$ is gene conversion rate, $f$ is proportion of fragmented mitochondria, and $t$ is the length of time over which gene conversion occurs;

iv. Every subsampling (for example, only allowing a subset of oDNAs to replicate) adds $n_c(1/n_2 - 1/n_1)$, where $n_1$ is initial copy number and $n_2$ is final copy number, and $n_c$ is sampled cluster size (see Box 1);

v. Every amplification (for example, doubling oDNA copy number after a cell division) adds $1/n_1 - 1/n_2$, where $n_1$ is initial copy number and $n_2$ is final copy number.

Eq 1 therefore shows that contributions from different mechanisms can contribute to beneficial oDNA segregation (we note in passing that (i) and (iv) contrast with a common binomial model of sampling, which is used sometimes to model the bottleneck but which assumes some unphysical behaviour; see Methods). Reference [16], for example, highlighted that the mouse mtDNA bottleneck can be achieved through a flexible combination of copy number depletion and mtDNA turnover, going some way towards reconciling competing observations and proposed mechanisms in the literature. Eq 1 extends this picture to capture the different balances between mechanisms that can give rise to a genetic bottleneck of a given size. The same amount of variance in heteroplasmy level can be generated by given amounts of subsampling, cell divisions, oDNA turnover, and/or gene conversion in combination.

To verify that Eq 1 accurately describes the contributions of different mechanisms to oDNA heteroplasmy level variance, we first conducted stochastic simulations of different mechanisms individually and in combination (see Methods) and confirmed good agreements between Eq 1 and these in silico results in each case (Fig 1B, S1 Fig).

## Theoretical predictions match single-cell heteroplasmy measurements with and without germline selection

To demonstrate the ability of our model to generate new mechanistic insight, we next turned to the well-characterised (but still debated) case of mtDNA in the mouse germline. Here, the timings of cell divisions have been measured, and mtDNA copy number has been recorded during germline development (see S1 Text and S4 Fig) [16,18,36,37], allowing quantitative prediction of heteroplasmy level variance contributions from different proposed mechanisms. To test these predictions, we used Eq 1 to predict the variance generated by the estimated 36 cell divisions in the mouse germline, with their timings and associated mtDNA population sizes taken from experimental observations (see Methods), and the ongoing turnover of mtDNA in the ageing germline. Divisions were paired with reamplification events to ensure that mtDNA copy number matched experimentally observed values throughout. The overall model thus comprised parts (i)+(ii)+(v) in the previous section and Box 1.

In Fig 1C and 1D, we verify that the predictions of Eq 1 match recent observations of heteroplasmy level variance in the mouse germline from reference [11] (using the "HB" model: a wild-derived mtDNA strain labelled HB admixed with C57BL/6N mtDNA on a nuclear C57BL/6N background). The overall model captures well the time behaviour of observed single-cell heteroplasmy level distributions (Fig 1C) and summary statistics (Fig 1D). We also used the Akaike information criterion (AIC) to compare models involving different combinations of variance-generating mechanisms, finding the most support for the (i)+(ii)+(v) model combining binomial partitioning at divisions with mtDNA turnover (Fig 1D, S2 Fig).

This new analysis of the HB mouse model agrees with mechanistic findings from an independent mouse model (admixed NZB and BALB/c mtDNA), while requiring none of the complicated mathematics in that earlier work [16]. Parameter inference for this model (see Methods) allows a quantitative estimation of the physical bottleneck size (the minimum number of mtDNA molecules per cell during development) of $670^{+200}_{-128}$, and an estimate for the combined statistic $vf/n$ of $1.4 \times 10^{-4} \pm 1.7 \times 10^{-5}$ day$^{-1}$, again agreeing well with the analysis of the NZB-BALB/c model, which estimated bottleneck size around 500 to 1,200 and turnover $v$ around 0.1 to 0.6 day$^{-1}$, corresponding to $v/n$ around $10^{-5}$ to $6 \times 10^{-4}$ [16]. This analysis also provides a quantitative demonstration of the fact that one variance-increasing mechanism can compensate for another in the animal germline. For example, if mtDNA turnover is lower, the variance generated by cell divisions is inferred to be higher (S2 Fig). In S2 Fig, we further demonstrate the use of Eq 1 to compare different hypothesised mechanisms (including subsampling and reamplification (iv)+(v) and partitioning of mtDNA clusters $n_c > 1$) to data on the mouse germline bottleneck using the NZB-BALB/c model [16].

Our theory also provides, to our knowledge, the first analytical description of heteroplasmy level variance under oDNA selection. Selective differences between mtDNA types are common, arising due to quality control or other mechanisms in the cell (reviewed in reference [24]), including in the animal germline [10,11]. Bias in gene conversion can also lead to effective selection of one oDNA type [23]. Selection leads to a change in mean heteroplasmy level over time, which complicates the analysis of heteroplasmy level variance. Despite this, our theory provides analytical descriptions of mean and variance behaviour in the general case of biased replication and/or gene conversion.

In Fig 1E and 1F and S1 Text, we demonstrate the use of this theory on another recent mouse model from reference [11]. In the "LE" model (wild-derived LE mtDNA admixed with C57BL/6N), selection was found to occur in concert with segregation in the germline. Application of our theory to this system again shows good agreement with the single-cell distributions of heteroplasmy level observed over time, with AIC values confirming that the model including selection performs significantly better than the model without selection for this case (Fig 1E) and allowing model comparison to be performed for mechanisms in this non-neutral case (Fig 1F). This is, to our knowledge, the first analytical theory predicting the joint effects of selection on heteroplasmy level mean and variance in the animal germline.

## Organelle fission–fusion state enhances and represses oDNA variance from different mechanisms

Having verified that our general model describes several specific instances of bottleneck mechanisms well, we next asked what specific mechanistic insights we could gain from its structure. Several diverse studies (including [4,12,30,51]) have suggested a role for fission–fusion dynamics of mitochondria in mtDNA segregation and maintenance. Reference [50] showed mathematically that the contribution to the variance from mtDNA turnover is scaled by $f$, the proportion of mtDNA molecules in fragmented mitochondria: $V'(h) = 2fvt/n$. This is because only fragmented mitochondrial elements are subject to turnover via autophagy [52]. Following this previous work, we investigated the hypothesis that organelle fragmentation may assist variance generation through fusion, but may also hinder the generation of variance via gene conversion (as recombination is limited by the physical separation of oDNA molecules). We used our framework to explore this hypothesis and seek a quantitative description of this physical influence on genetic dynamics.

Our model generalises the expression from reference [50] to $V'(h) = (1+n_d)fvt/n$, in the case where $n_d$ oDNA molecules are removed by an autophagic event (we recover the result from reference [50] when $n_d = 1$, as assumed therein). Further, we found that the variance contribution from gene conversion also takes a simple form: $V'(h) = 2g^2\kappa t$, where $g$ is the proportion of oDNAs available for recombination. In a simple picture of the cell, consisting of fragmented and networked organelles, $g = (1-f)$. Hence, variance generated by turnover scales with $f$ and that generated by gene conversion scales with $(1-f)^2$, illustrating the hypothesised tension above (Fig 1B inset). This simple result means that the cell can control the mechanism of oDNA segregation by controlling organelle dynamics (as well as the expression of factors involved in each process). In a highly fused network, any segregation must occur through recombination, as turnover is relatively limited. In a fragmented population, segregation via recombination can only occur through limited "kiss-and-run" events (transient interactions where individual organelles physically meet, potentially exchange oDNA, and separate on short timescales) [53].

The analysis above assumes a "mitochondrial" picture where fission and fusion mix the oDNA content of the chondriome. In a "plastid" picture, organelle fusion and thus mixing of genetic content is rare. In this situation, organelles can be pictured as independent containers of oDNA, with cellular heteroplasmy made up of the aggregate oDNA content throughout all these containers. We show in S1 Text that the consequence of this separation is a rescaling of gene conversion rate $\kappa$: The rate is divided by the number of individual, separated organelles, reflecting the fact that less diversity can be generated through recombination in a set of smaller gene pools than in one large pool. This limitation does not remove the capacity of gene conversion to segregate ptDNA, however, and depending on ptDNA and plastid numbers, the limitation may be of rather low magnitude (see S1 Text).

The form of organelles also impacts the distribution of oDNA molecules at cell division. If oDNA molecules are randomly distributed throughout the cell and a division plane bisects the

cell, the partitioning of oDNA to each daughter is binomial. If the oDNA molecules are perfectly separated with exactly half either side of the division plane, the partitioning will be "perfect." Active spread of organelles through the cell can reduce the variance of this partitioning to an intermediate case between binomial and perfect partitioning [46,54]. This spreading could, for example, be achieved through repulsion of individual organelles, or the formation of a space-filling network prior to division [54,55]. We used a simple physical simulation to show that these partitioning mechanisms impact the copy number statistics, but not the heteroplasmy level variance, across daughter cells (S3 Fig).

### Gene conversion increases variance independent of copy number

Each of the above mechanisms (i) to (v) (Eq 1) contributes a given amount of cell-to-cell heteroplasmy level variance to a cellular population. The magnitudes of most of these contributions are divided by the copy number of oDNA (for example, a cell division adds $n_c/n$ to the variance, where $n$ is copy number prior to division). This means that variance is harder to increase in large oDNA populations. As oDNA copy numbers are typically hundreds or thousands per cell, several of these mechanisms must be applied many times to increase variance to useful levels. The exception is the contribution of gene conversion. Gene conversion at rate $\kappa$ contributes $2\kappa$ to the variance per unit time regardless of the size of the oDNA population ((iii) in Box 1; asterisk in Fig 1B).

This lack of copy number dependence potentially makes gene conversion a powerful mechanism for increasing variance in cases where oDNA population size cannot readily be reduced. In mammals and other animals, a germline sequestered early in development provides a compartment for mtDNA copy number depletion. Multicellular organisms without a fixed body plan do not typically sequester a germline [21], and unicellular organisms cannot sequester a germline in the same sense (although ciliates partition nuclear genetic information into less active and more active compartments within the cell [56]). In these cases, organelles are inherited from cells that must presumably retain bioenergetic function through much of the previous generation's life. oDNA depletion may therefore not be an option. The limited observations that exist of oDNA copy number during very early plant development, for example, suggest much more limited copy number change than in animals, and indeed point to an increase rather than decrease at early developmental stages (S1 Text, S4 Fig), and subsequent oDNA copy number change, while debated, also seems limited [57]. This lack of a physical bottleneck limits the extent to which variability can be induced by other mechanisms, but gene conversion is not limited in this way. We therefore hypothesised that organisms without an early-sequestered germline would be more likely to utilise gene conversion to generate beneficial oDNA variance [23].

### mtDNA recombination and variance generation in organisms without an early-sequestered germline

To test this hypothesis, we next turned to systems biology to characterise the cases in which gene conversion may influence oDNA variance, focusing first on mtDNA. We sought evidence for a link between presence of genes known to be involved in mtDNA recombination surveillance and the absence of a dedicated germline across eukaryotes [58]. For this cross-kingdom analysis, we particularly focused on the genes *msh1*, *mgm101*, and *mhr1*, functioning in mtDNA surveillance and highlighted as present across diverse taxa in reference [26].

We begin by noting that these observations must be interpreted with the phylogenetic relationship of modern species in mind; it is not the case that species constitute independent observations of any given pattern. We found that multicellular lineages lacking a fixed body plan (including corals, sponges, anemones, sea pens, plants, algae, and fungi) typically possess one or more of these mtDNA recombination surveillance genes, which organisms with a fixed

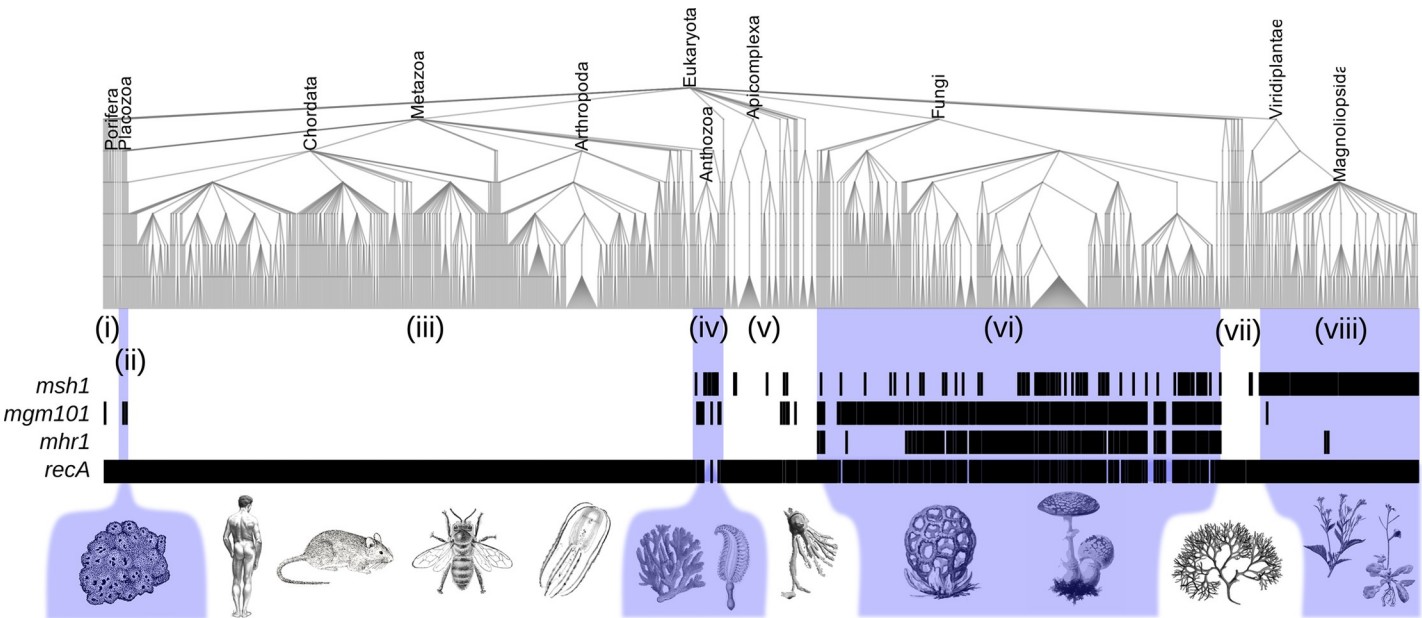

**Fig 2. Organelle recombination genes and body plans across taxa.** Presence (black) or absence (white) of recombination genes *msh1*, *mgm101*, *mhr1*, and *recA* across eukaryotes. Metazoa in (iii) are those with fixed body plans that typically sequester a dedicated germline in early development; none of these species encode *msh1*, *mgm101*, or *mhr1*. Metazoa without fixed body plans (including Porifera and Placozoa (ii) and Anthozoa (iv)), as well as other kingdoms without animal-like germlines ((i) and (v)–(viii)), typically encode oDNA recombination surveillance machinery. A simplified version (truncated at higher taxonomic levels) and individual gene trees are in S5 Fig; for data and code, see https://github.com/StochasticBiology/odna-segregation.

body plan completely lack (Fig 2; following numerals refer to clades within this figure; also see S5 Fig for more details). *msh1*, *mgm101*, and *mhr1* were universally absent across chordates and arthropods (iii). In plants (viii), *msh1* is ubiquitous; some fungi (vi) also possess *msh1* but most also possess *mgm101*, with many also possessing *mhr1* [26,59]. oDNA structures in these lineages exhibit pronounced structural diversity, including branched molecules, molecules of different sizes, reorganisations, and other features concomitant with recombination [26,60]. Brown and red algae ((i) and (vii)) with broadly plant-like forms also seem to possess *msh1*, suggested by moderate BLAST hits in model species *Ectocarpus siliculosis* and *Chondrus crispus*, respectively, among others (S5 Fig). Several unicellular eukaryotes ((i), (v), and (vii)) also encode *msh1* and *mgm101*, including alveolates, amoebas, and slime moulds [59].

Strikingly, several examples also exist of metazoan lineages without fixed body plans that have acquired organelle recombination genes. The famous cases of corals, some of which have acquired a form of *msh1* in their mitochondrial genomes [61,62], fit this pattern (iv). Structural diversity in coral mtDNA, including the presence of introns and evidence for inversion and reorganisation events [62], supports the capacity for recombination. Some placozoans, sponges, and cnidarians also encode *mgm101* (ii) [59]. By contrast, ctenophores, also considered basal metazoa (or a sister clade to cnidarians), lack recombination genes, but appear to have more fixed body plans with a more sequestered germline (iii) [58,63].

## Genetic and physical features facilitate variance-increasing oDNA recombination in organisms without animal-like germlines

To pursue this hypothesis that gene conversion may induce beneficial heteroplasmy level variance without requiring an animal-like bottleneck, we next asked to what extent "germline" oDNA gene conversion is made possible by organelle ultrastructure and gene expression

patterns in non-metazoan organisms. To this end, we turned first to *Arabidopsis thaliana*, probably the most studied multicellular organism without a fixed body plan. Plant mitochondria are usually fragmented [25,31], which according to Eq 1 would appear to prevent any pronounced increase in variance due to this facilitation of gene conversion. However, observations of mitochondrial dynamics in *Arabidopsis* [32] show that a highly fused network state, usually rare in plants, occurs in shoot apical meristem (SAM; the organ that gives rise to the aboveground germline), but not the belowground root meristem (Fig 3A). Analysis of the microscopy data from reference [32] suggests that in cells in the SAM, only around 30% of the mitochondrial mass in a cell may be in a physically fragmented state, with an associated dramatic increase in potential recombination (which, as before, scales as $(1-f)^2$, where $f$ is the fragmented proportion). This fused state has indeed been hypothesised to facilitate recombination to ensure the inheritance of a homogeneous mtDNA population [32]. Our theory agrees in the sense that more cell-to-cell diversity in the SAM makes the probability of a homogenous (homoplasmic wild type or mutant) cell more likely.

However, an increased potential capacity for oDNA recombination in cells that will give rise to the germline is not itself sufficient to facilitate gene conversion. mtDNA recombination

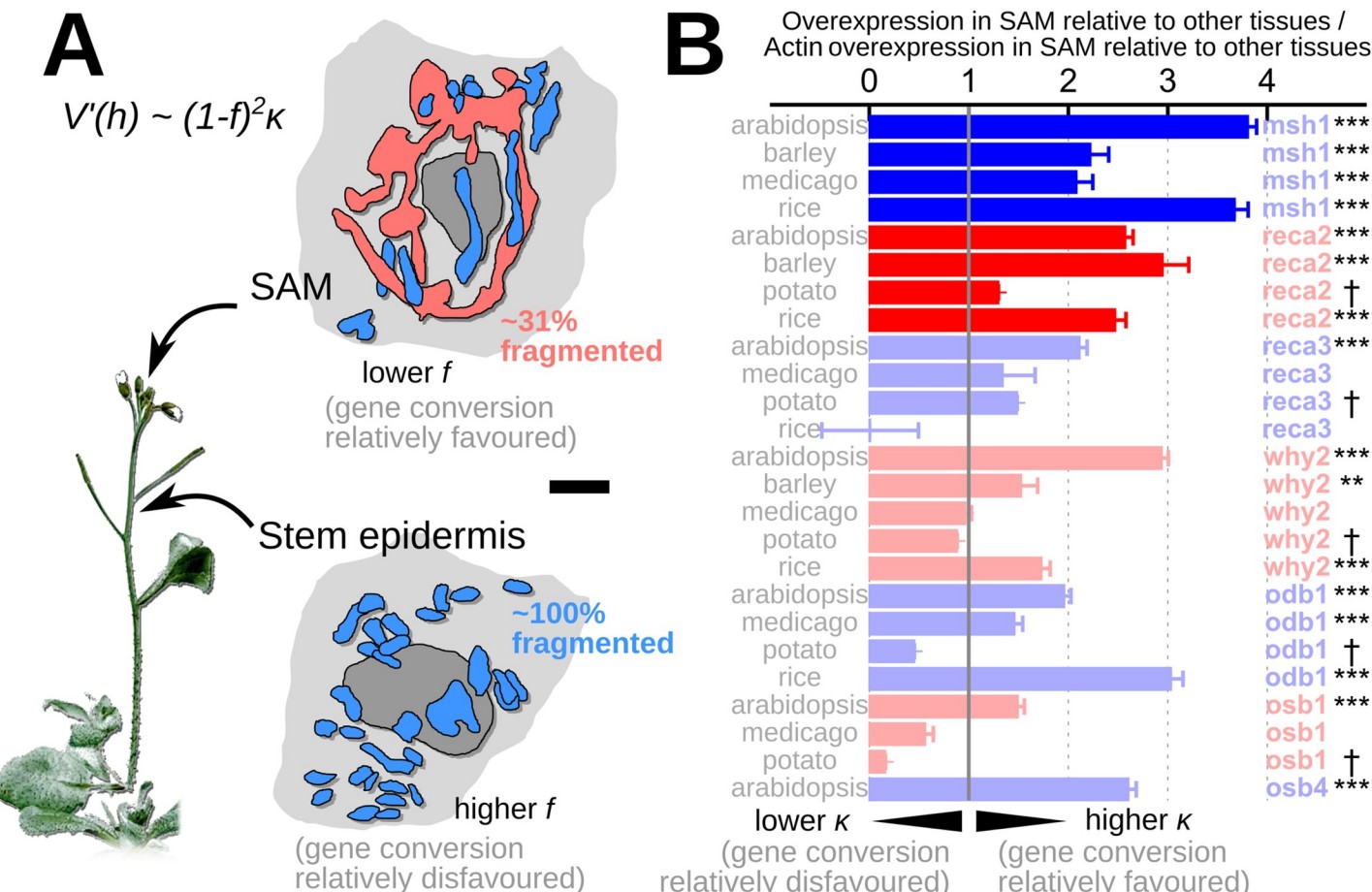

**Fig 3. Physical and genetic features facilitating germline gene conversion across plant species and tissues.** (A) Single-cell mitochondrial ultrastructure in SAM and epidermis of *Arabidopsis*, from image data in reference [32]. Nucleus in grey; mitochondria in blue; single fused cage-like mitochondrion in red. This large-scale fusion decreases *f*, the proportion of mtDNA molecules in fragmented organelles, and facilitates recombination. Scale bar is 2 *μ*m. (B) Tissue-specific recombination surveillance gene expression across plant species. Values give the overexpression in SAM relative to other tissues, normalised by the SAM overexpression of actin relative to other tissues. *msh1* and *reca2* are highlighted as foci in the main text; other genes are also associated with modes of recombination favouring gene conversion [48]. ***, *p*<0.001; **, *p*<0.01; *, *p*<0.05, after Bonferroni correction; †, insufficient data for variability analysis; for data and code, see https://github.com/StochasticBiology/odna-segregation. SAM, shoot apical meristem.

underlies several processes in *Arabidopsis*, and more broadly in plants, with the relative activity of these processes modulated by several surveillance factors including RECA2, RECA3, MSH1, OSB1, and OSB4 (reviewed in [48]). These factors repress RecA-independent recombination, a damage recovery pathway that is competitive with RecA-dependent D-loop formation, a precursor to gene conversion. Other factors including ODB1 and WHY2 promote this RecA-dependent pathway. Hence, the expression of these genes favours gene conversion over other recombination-mediated events. We hypothesised that these genes would be expressed highly, hence relatively promoting gene conversion, in tissues that contribute to the inheritance of mtDNA.

Following this hypothesis, we asked where these genes are expressed in the plant. Using a tissue-specific transcriptomic atlas of *Arabidopsis* genes [64], we found that most of these factors are substantially and significantly more highly expressed in the SAM than other tissues (Fig 3B). These expression patterns appear conserved across other plants where corresponding gene expression maps are available, including rice, barley, potato, and *Medicago trunculata*. Taken together, these observations support a picture in the SAM where mitochondrial dynamics increase recombination capacity, and the parallel expression of recombination surveillance genes shifts recombination poise to favour gene conversion. As the SAM will ultimately produce the next generation's sex cells, this combined physical and genetic control provides the capacity for generating heteroplasmy level variance without depleting oDNA copy numbers. Interestingly, during preparation of this report, very recent observations were published of the expression of another recombination factor, MOC1, in algae and bryophytes [65], which shows similar behaviour and may qualitatively support this hypothesis across a broader taxonomic range.

Eq 1 predicts that this increased fusion (decreasing $f$) coupled with a shift in the balance of recombination activity towards gene conversion (increasing $\kappa$) will increase the possible contribution to heteroplasmy level variance from recombination. The coupling of physical and genetic changes in the SAM thus provides potential foundations for beneficial oDNA segregation in what will become the germline in plant species.

## Discussion

We have quantitatively shown how different cellular and subcellular mechanisms can contribute to the increase of heteroplasmy level variance, and propose that different eukaryotic taxa use different combinations of these mechanisms (Fig 4). This beneficial cell-to-cell variability provides a substrate upon which purifying selection can act, circumventing Muller's ratchet and preventing mutational meltdown of organelle populations. This theory can be applied across eukaryotes—both providing new quantitative insights on the animal germline bottleneck and describing mechanisms supporting oDNA segregation in nonanimal eukaryotes.

In particular, many metazoans that have evolved an early-sequestered germline can use mechanisms which harness the depletion of cellular oDNA populations. Other organisms without a fixed body plan can use recombination-mediated processes that generate variance independently of oDNA population size, with gene conversion providing an "alternative" bottleneck. This theory is supported by cross-taxa observations of recombination surveillance genes supporting gene conversion across diverse lineages without fixed body plans, and not within animal lineages that sequester an early germline, and by specific facilitative ultrastructural and gene expression features observed in the SAM of plants. This picture provides an explanatory link between diverse organism physiologies and organelle behaviours observed across taxa and provides what we believe is the first comprehensive quantitative framework with which to understand these mechanisms.

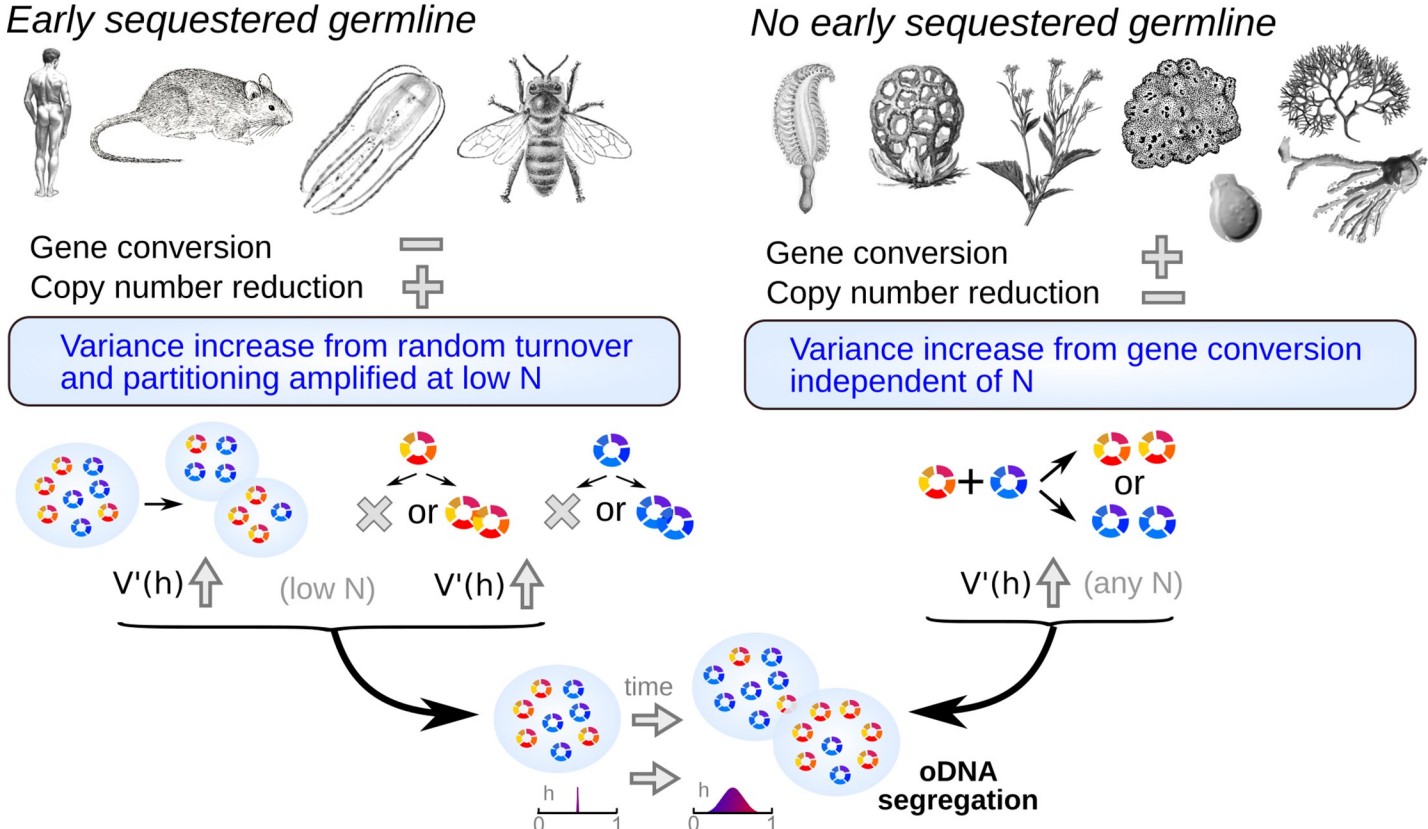

**Fig 4. Proposed differences in mechanisms for oDNA segregation across eukaryotic taxa.** Early germline sequestration allows copy number reduction and a "physical bottleneck"; otherwise, recombination-mediated gene conversion can increase oDNA variance. oDNA, organelle DNA.

The mathematical aspects of our theory rely on several approximations. The analytical techniques we use work well for circumstances that correspond to moderate heteroplasmy level variance. Heteroplasmy level variance cannot continue to increase unbound, as the repeated action of the terms in Eq 1 would suggest. Instead, it approaches a maximum value (corresponding to homoplasmy in all cells), and its increase slows as it nears this point. Characterising this slowing is possible within the framework we use, but requires a more involved analysis [66] (or numerical simulation). We present the first-order picture as it describes the magnitudes of effects observed in most biological systems considered (reviewed in reference [14]) and refer the reader to S1 Text for more details. Further, the model we use relies on a particular (Poissonian) "memoryless" representation of these subcellular processes, where the probability of an event occurring only depends on the current state of the system. This is a common and successful paradigm in quantitative modelling of cell biology but cannot be a perfect microscopic description of these processes [67]. It is unlikely that the results of a more precise model accounting for, for example, delays and memory effects in these processes, would yield substantially different results, but such an analysis would be a desirable future extension. Finally, we do not consider mutational processes that change one oDNA type into another. While our mathematical approach can naturally incorporate such de novo mutation, it complicates the resulting expressions and requires further analysis, which is the target of ongoing development. The model therefore will not capture the full dynamics in situations where oDNA mutation occurs as or more frequently than the other processes we model. The absence of de novo variant generation means that, in principle, the processes described in our model can be

applied to any mutational variant (for example, structural variants) in addition to variants differing by base pair changes. However, caution should be applied when interpreting gene conversion processes in this case, as it is not clear that our model gene conversion process applies to the interaction between structural variants, where more detailed modelling of recombination processes may be required [68,69].

Given the power of gene conversion to increase oDNA variance, why is its use limited? The answer may be that oDNA recombination raises potential issues as well as generating beneficial variance. Rearrangements of the oDNA genome raise the possibility of structurally compromised oDNA molecules arising (and proliferating) within organelles and cells. This may involve fragmented oDNA molecules, encoding only a subset of genes, corresponding to the "minicircle" versus "master circle" (complete oDNA molecules) picture in plants [31,51,70] (which is not unchallenged [71]). Shifts in the relative balance of these structurally different oDNA types (as in substoichiometric shifting, influenced by recombination genes including *msh1* [25,28,29,72]) may challenge cellular bioenergetics, particularly if fragmented molecules experience a replicative advantage. Such molecules, replicating more and contributing less to bioenergetic poise, correspond to selfish genetic elements [25,42] (see S1 Text). The animal strategy of early sequestration of a dedicated germline avoids this potential complication by generating variance without recombination.

In yeast, where mitochondria are highly fused and recombination then less physically limited, the presence of selfish elements including the well-known "petite" mutant is observed ubiquitously [73,74]. Our theory suggests that similar selfish molecules may emerge in organisms that leverage recombination-mediated processes to generate oDNA diversity and maintain a physically well-mixed pool of oDNA molecules (S1 Text) [30]. The general fragmentation of mitochondria in plants (except in the SAM where diversity generation may be the key priority) would guard against this emergence by limiting recombination in nongermline tissues. This picture is supported by the high expression of *msh1* in the SAM of various plants, where fusion allows more recombination activity.

In plants [75], corals [62,76], sponges [20], and fungi [77], mtDNA mutation rates (assessed via a range of specific quantities) have long been observed to be lower than nuclear mutation rates. This observation is in contrast with other metazoans, where mtDNA mutation rates are higher than nuclear mutation rates [77]. Recombination-mediated damage repair [48,60] is a likely contributor to these observations. Our theory describes the influence of gene conversion on oDNA populations; other recombination mechanisms acting preferentially to favour one oDNA type over another can readily be analysed by including the corresponding processes in our model (see S1 Text).

## Methods

### Individual oDNA dynamics

We consider a cell with oDNA state described by copy numbers $(W_f, W_s, M_f, M_s)$, where $W$ is wild type, $M$ is mutant, subscript $f$ denotes oDNA molecules in a fused organelle, and subscript $s$ denotes molecules in a "singleton" fragmented organelle [50]. Total copy number $N = W_f + M_f + M_s + W_s$, heteroplasmy level $h = (M_f + M_s)/N$, and the statistic $f = (W_s + M_s)/N$ describes the proportion of oDNAs in a fragmented state. We allow several different coarse-grained representations of the dynamic organelle population into fused and fragmented parts and show that our results do not depend on these choices (see S1 Text). We model oDNA as undergoing relaxed replication with rate $\lambda$ = constant × $(1 + \alpha N)$, applying feedback to drive the system towards a desired total copy number $\alpha^{-1}$. A set of Poisson processes (rates in parenthesis) govern replication ($\lambda$), degradation ($\nu$) and gene conversion ($\kappa$) of oDNAs, and fission ($\alpha_s$) and

fusion ($\alpha_f$) of organelles (see S1 Text). We use the Kramers–Moyal expansion to construct a Fokker–Planck equation as above, then use Itō's formula to recast this as a Fokker–Planck equation for heteroplasmy level $h = (M_f + M_s)/N$ [50,67]. We then extract ODEs for the drift and diffusion terms for heteroplasmy level from this Fokker–Planck equation and solve them to obtain time behaviour of $E(h)$ and $V(h)$.

We include several parameters allowing a general picture of this system: $\delta$ is a replicative difference between wild-type and mutant mtDNA; $\epsilon$ is gene conversion bias; and $n_d$ is the characteristic number of oDNAs that are degraded by an autophagic event.

## Selection

We assume that selection can be manifest through a difference in replication rates $\delta$ or a bias $\epsilon$ in gene conversion towards one mtDNA type. In the case of selection, $h$, $f$, and $N$ can generally change over time. We assume that physical dynamics occur on a faster timescale than genetic dynamics [50] and treat $f$ and $N$ as equilibrated constants. In S1 Text, we obtain general expressions for $E(h)$ and $V(h)$ under biased replication and gene conversion. For the mouse example in the main text, we ignore recombination and assume large copy numbers, and can therefore use the simpler results

$$E(h) = \frac{1}{1 + e^{-\rho t}} \qquad (2)$$

$$V(h) = \frac{\exp\left(\frac{-2}{1+e^{\rho t}}\right)}{4(e^{\rho t} + 1)N\rho} \left( 4evf + 4vfe^{\rho t+1} + \exp\left(\frac{2}{e^{\rho t} + 1} + \rho t\right)(\rho - 4vf) - \exp\left(\frac{2}{e^{\rho t} + 1}\right)(\rho + 4vf) \right), (3)$$

where $\rho = \delta(\alpha N - 1)$ is the scaled replicative difference between mtDNA types. These expressions are useful in comparison with experimental heteroplasmy observations, as turnover $vf$, selection $\rho$, and copy number $N$ are the only variable combinations required to be independently identified.

## Sampling and amplification

Subsampling without replacement is modelled via the hypergeometric distribution, although we also consider subsampling with replacement via the binomial distribution. Amplification is modelled with a Pólya urn, giving rise to a beta-binomial distribution (see S1 Text).

We use a hypergeometric model rather than a binomial model for sampling and cell divisions. In binomial sampling, oDNA molecules are sampled with replacement, so that the same molecule can be sampled several times over. Under a binomial model, contributions (i) and (ii) are $2/n$ and $1/n_2$, respectively. However, this picture leads to unrealistic behaviour such as allowing several copies of the same molecule to be inherited by a daughter cell or the same molecule to be inherited by both daughter cells. The forms for (i) and (ii) above instead model sampling without replacement, so each molecule has only one possible fate. This involves replacing the binomial model with a hypergeometric model (see S1 Text).

## Total variance

We use a linear noise approximation and assume that different contributions to heteroplasmy level variance can be summed. We work with normalised heteroplasmy level variance

$$V'(h) = \frac{V(h)}{E(h)(1 - E(h))} \qquad (4)$$

## Stochastic simulation

To check theoretical results, we use Gillespie's stochastic simulation algorithm [78] of our stochastic system, with $10^4$ instances and default parameterization $\lambda = 2$, $\nu = 1$, $\alpha = 1/1000$, $\kappa = 0.002$, $n_d = 1$. We simulated many variations of this default set to assess the model's ability to describe different processes (see S1 Text). Confidence intervals on $V'(h)$ from simulation were estimated using the estimated standard error of the variance (under a normal assumption, which is not unreasonable for our simulations performed at intermediate heteroplasmy) of $V'(h)\sqrt{2/(n-1)}$ [14].

## Mouse germline

The development of the female germline is estimated to involve 29 divisions with a period of 7 hours, during which copy number is depleted from around $10^5$ to a debated number, 7 divisions of period 16 hours, during which copy number is reamplified to around 5,000, and subsequent ongoing reamplification to the original $10^5$ [16]. In S1 Text, we use Eq 1 to show that, if copy number changes from $N_0$ to $N_k$ over $k$ cell cycles, $V'(h) = (\alpha+n_c-1)/(\alpha N_0)((2/\alpha)^k-1)((2/\alpha)-1)$, where $\alpha = 2(N_k/N_0)^{1/k}$. In the absence of recombination and selective differences between mtDNA types, the only remaining variance contribution is from mtDNA turnover [16] (or subset replication [37]).

 We used a maximum likelihood approach with bootstrapping to infer parameter estimates and confidence intervals for models involving different combinations of these features. Specifically, $10^3$ bootstrap resamples were used, and optimisation was performed in R [79] with *optim*, by default implementing the Nelder–Mead algorithm [80]. The observed data were the individual heteroplasmy level measurements of the (neutral) HB and (non-neutral) LE models from reference [11]. Heteroplasmy level was transformed, to account for the expected differences in dynamics given different initial conditions, using $\delta = g(h; h_0) = \log\left(\frac{h(h_0-1)}{h_0(h-1)}\right)$, where $h$ is heteroplasmy level, and $h_0$ is initial or reference heteroplasmy level [24]. When an initial heteroplasmy is not defined, we use $h_0 = \frac{1}{2}$ as a reference value. We use the delta method to account for the transformation in model fitting, as described in S1 Text. We used the AIC and $R^2$ values to compare models.

## Genomic data

We used NCBI's Gene tool (https://www.ncbi.nlm.nih.gov/gene/) to identify annotated accessions where our recombination surveillance genes of interest were present. Manual curation removed some false positives (including, for example, misidentified *msh2* in humans and muscle-specific homeodomain 1 in *Drosophila*). We complemented this approach with specific BLAST analyses. Specifically, we performed blastx searches against the nonredundant protein sequence database nr, using as queries NM_113339.4 (*Arabidopsis* nuclear-encoded *msh1*), NC_036022.1 (6348..9287) (*Dendronepthya* mtDNA-encoded *msh1*), NC_001140.6 (349574..352453) (*Saccharomyces cerevisiae* S288C *msh1*), NC_001142.9 (700882..701691) (*S. cerevisiae* S288C *mgm101*), and NC_001136.10 (1055212..1055892) (*S. cerevisiae* S288C *mhr1*). We used an E-value threshold of 1 except for the *msh1* queries where we imposed an E-value threshold of $10^{-50}$ for the plant and coral searches and $10^{-100}$ for the fungal search, chosen from preliminary investigation to avoid hits from the similar members of the *msh[X]* family. We used the Common Taxonomy Tree tool (https://www.ncbi.nlm.nih.gov/Taxonomy/CommonTree/wwwcmt.cgi) to embed the corresponding species on an illustrative taxonomy.

### Tissue-specific gene expression data

We used the University of Toronto's Bio-Analytic Resource for Plant Biology (BAR; [81]) to compile tissue-specific gene expression data from Schmid and colleagues [64]. The compiled data are normalised by BAR using the Affymetrix GeneChip Operating Software (GCOS) method with a target intensity (TGT) value of 100 [81]. To account for possible general high (or low) levels of gene expression in the SAM, we compared the level of SAM overexpression of each gene of interest with the level of SAM overexpression of a control gene (actin). We computed 4 quantities: normalised expression level of gene of interest in SAM ($G_{SAM}$), normalised expression level of gene of interest in other tissues ($G_{other}$), normalised expression level of control gene in SAM ($C_{SAM}$), and normalised expression level of control gene in other tissues ($C_{other}$). We then computed the relative SAM abundance of the gene of interest, normalised by the relative SAM abundance of the control gene: $((G_{SAM}/G_{other})/(C_{SAM}/C_{other}))$. Standard uncertainty propagation was used to compute the associated standard deviation (most samples are triplicated).

## Supporting information

**S1 Text. Stochastic modelling of oDNA populations.** Stochastic modelling of physical and genetic oDNA dynamics at and between cell divisions and summary of existing data on oDNA during development. oDNA, organelle DNA.
(PDF)

**S1 Fig. Time series of heteroplasmy level statistics.** Theory and stochastic simulations ($10^4$ repeats; error bars—often small compared to the point labels—give 95% confidence intervals) of heteroplasmy level variance $V'(h)$ as a function of time for different mechanisms. Labels give (division)/(reamplification) dynamics. Division can be deterministic (DD), hypergeometric (HD[$n_c$]), or binomial (BD[$n_c$]), where $n_c$ is cluster size (or individual molecules if absent). Reamplification can be deterministic (DA) or random (RA). The three panels correspond to different post-division population size $n_2$, reflecting either halving ($n_2 = 500$) or more budding-like divisions (lower/higher $n_2$).
(TIF)

**S2 Fig. Model comparison for different theories for the mouse germline bottleneck. (A–C)** Distributional and variance predictions, following Fig 1 C–F, for different combinations of (i) binomial cell divisions, (ii) mtDNA turnover, (iv) mtDNA subsampling, and (v) mtDNA reamplification. In each case, the different models are fitted to data. (A) Application of neutral variants of model to data from the HB model of reference [11], following Fig 1C. (B) Application of non-neutral variants of model to data from the LE model of reference [11], following Fig 1E. (C) Application of neutral variants of the model to the NZB-BALB/C model from references [37,36,18] (following reference [16]). (i)+(v) contribute early variance, of magnitude $V_0$, during the developmental bottleneck; (ii) contributes ongoing variance increase at rate $2vf/n$. (iv), allowing only a proportion of mtDNA molecules to replicate, can potentially contribute variance over different timescales; here, we illustrate it as a single discrete event during oogenesis [37], but other instances give contributions comparable to the (ii) model [16]. **(D–F)** Bootstrapped distributions for the (i)+(ii)+(v) model and the HB model, for (D) $vf/n$, (E) $V_0$, and (F) both variables. As the copy number dynamics and timing of early mouse development have been well characterised, a given $V_0$ value can be interpreted as a value for $b$, the minimum mtDNA copy number during development; we present example values on the upper horizontal axis. (F) shows that a low value from 1 variance contribution can be compensated by a high

value from the other. mtDNA, mitochondrial DNA.
(TIF)

**S3 Fig. Partitioning at cell divisions.** (Top) Example snapshots of the interacting and noninteracting simulations. (Bottom) Variance of copy number *N* and heteroplasmy level *h* for each case.
(TIF)

**S4 Fig. mtDNA copy number in the germline.** Measurements from **(A)** mouse models and **(B)** sparser measurements from different plants (see text), where arrows link observations across development in the same species from the same study. *, mtDNA copy number in rice estimated via copy number of individual mtDNA genes. Question marks denote averages for which uncertainty is not immediately available from the source publication. mtDNA, mitochondrial DNA.
(TIF)

**S5 Fig. Recombination surveillance genes across taxa. (A)** Reduced taxonomic tree, corresponding to an averaging of gene presence over leaves in Fig 3. Grayscale rectangles give the proportion of leaves under each parent node that were found to contain the given gene (white, none; black, all). **(B)** Taxonomic trees linking BLAST hits for *msh1*, *mgm101*, and *mhr1* across eukaryotes. Anthozoan species are highly sampled for *msh1* because the gene is present in the mitochondrial genome and hence historically easier to characterise; the plant group in the figure reflects the vast majority of annotated plant genomes. Groups labelled include soft corals (Alcyonacea), blue corals (within Helioporacea), and sea pens (Pennatulacea). *mgm101* is ubiquitous in fungi; in addition to those fungal families labelled, we also find hits in placozoans (Placozoa), sea anemones (Actiniaria), sponges (Porifera), corals (Anthozoa), slime moulds (Dictyosteliales), and other protists (for example, Physariida). *mhr1* is more limited to ascomycete fungi.
(TIF)

## Acknowledgments

The authors are grateful to Professor Ralph Bock for valuable comments on the manuscript and to Paul Una for useful discussions. We are also grateful to the creators of the public domain images of different eukaryotes used in Figs 2–4.

## Author Contributions

**Conceptualization:** Ellen C. Røyrvik, Iain G. Johnston.

**Data curation:** Joanna M. Chustecki, Konstantinos Giannakis, Robert C. Glastad, Arunas L. Radzvilavicius, Iain G. Johnston.

**Formal analysis:** David M. Edwards, Arunas L. Radzvilavicius, Iain G. Johnston.

**Funding acquisition:** Iain G. Johnston.

**Investigation:** David M. Edwards, Ellen C. Røyrvik, Joanna M. Chustecki, Konstantinos Giannakis, Robert C. Glastad, Arunas L. Radzvilavicius, Iain G. Johnston.

**Methodology:** David M. Edwards, Ellen C. Røyrvik, Iain G. Johnston.

**Project administration:** Iain G. Johnston.

**Software:** Iain G. Johnston.

**Supervision:** Iain G. Johnston.

**Visualization:** Iain G. Johnston.

**Writing – original draft:** Iain G. Johnston.

**Writing – review & editing:** David M. Edwards, Ellen C. Røyrvik, Arunas L. Radzvilavicius, Iain G. Johnston.

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
