## [Editor Report · Decision Letter 0]

11 Dec 2020

Dear Dr Johnston, 

Thank you for submitting your manuscript entitled "Avoiding organelle mutational meltdown with or without a germline bottleneck" for consideration as a Research Article by PLOS Biology.

Your manuscript has now been evaluated by the PLOS Biology editorial staff, as well as by an academic editor with relevant expertise, and I'm writing to let you know that we would like to send your submission out for external peer review.

IMPORTANT: The Academic Editor was somewhat equivocal in their advice, identifying a potential problem with the current layout of your manuscript. I'll paste in their comments verbatim, and we ask that you try to remedy these issues upfront before we send it out to review, as we think this will give your manuscript the best chance in the process.

Academic Editor's comments:

"...for a work of this nature, I feel that it's not ideal to have the real core of the hypothesis relegated to the rather challenging Supplementary Information. I would prefer to see the framework of hypotheses clearly laid out within the main text, followed by details of the quantitative modelling, together with a careful description of the strengths and limitations of the different elements of the models and of their robustness. Given this, I'm not sure that PLOS Biology is the best destination for the manuscript. A specialist journal might well be more appropriate."

If possible, we would like you to achieve some of this adjustment, but without compromising the accessibility of the paper to our broader readership (we admit that this is quite a tall order)!. In addition...

Please re-submit your manuscript within two working days, i.e. by Dec 13 2020 11:59PM.

Kind regards,

Roli Roberts

Senior Editor

PLOS Biology

---

## [Decision Letter · Decision Letter 1]

2 Feb 2021

Dear Dr Johnston,

Thank you very much for submitting your manuscript "Avoiding organelle mutational meltdown with or without a germline bottleneck" for consideration as a Research Article by PLOS Biology. As with all papers reviewed by the journal, yours was evaluated by the PLOS Biology editors as well as by an Academic Editor with relevant expertise and in this case by three independent reviewers.

You'll see that all three reviewers are very positive about your study. However, they each make a number of requests which should be attended to before further consideration.

Based on the reviews, we will probably accept this manuscript for publication, assuming that you will modify the manuscript to address the remaining points raised by the reviewers. Please also make sure to address the data and other policy-related requests noted at the end of this email.

IMPORTANT:

a) Please could you modify your title slightly to emphasise the broad scope of your study: "Avoiding organelle mutational meltdown across eukaryotes with or without a germline bottleneck"

b) Please attend to all reviewer requests.

c) Please attend to my Data policy Requests further down this email.

We expect to receive your revised manuscript within two weeks. Your revisions should address the specific points made by each reviewer. 

-  a cover letter that should detail your responses to any editorial requests, if applicable

*Published Peer Review History*

*Early Version*

Sincerely,

Roli Roberts

Senior Editor,

rroberts@plos.org,

PLOS Biology

DATA POLICY:

Regardless of the method selected, please ensure that you provide the individual numerical values that underlie the summary data displayed in the following figure panels as they are essential for readers to assess your analysis and to reproduce it: Figs 1BCDE, 2, 3B, S1, S2ABCDEF, S3, S4, S5. I note that you are planning a GitHub deposition, so you could include these data there or as supplementary data files attached to the manuscript. NOTE: the numerical data provided should include all replicates AND the way in which the plotted mean and errors were derived (it should not present only the mean/average values).

REVIEWERS' COMMENTS:

Reviewer #1:

The genomes of mitochondria and chloroplasts are passed from generation to generation via only one of the parents, which makes organelle DNA susceptible to Muller's ratchet. The question is then, how can organisms avoid the gradual accumulation of mutations in their organelles? 

The authors constructed an analytical mathematical model that investigates the effects of various mechanisms like a germline bottleneck, gene conversion, mitochondrial dynamics and cell division on the ability to generate cell-to-cell variance in the level of heteroplasmy. This variance can then be used by selection processes to prune cells carrying a high mutational load.

This work studies an important question and their mathematical model provides surprisingly simple expressions for the contributions of different biological mechanisms to cell-to-cell heteroplasmy level variance (Box 1). One of the main results is that gene conversion acts independently of organelle copy number. This explains how organisms without organelle bottleneck (like plants) can avoid mutation accumulation. This insight nicely explains and unifies different organelle structure and behaviour across different taxa. 

The analytical model is complemented by stochastic simulations and analysis of different experimental data sets that corroborate and support the analytical results. I very much like this combination of approaches and have only a few minor suggestions to make.

In Fig 1B the bars show analytical predictions and the white points results from stochastic simulations. Is that the mean of 5000 repetitions, as mentioned in line 429? This should be mentioned in the figure legend and maybe it would be useful to add 95% confidence intervals. 

In Fig 1CDE what is the meaning of 'Transformed h' and 'dpc' ?

In Fig S1 I wonder again about the data points for the numerical simulations. How many repetitions have been used and what is the CI ?

In summary, this paper studies an important questions regarding the inheritance of organelle DNA and provides interesting and elegant answers. I therefore like to recommend it for publication after minor modifications. 

Reviewer #2:

High cell-to-cell variance in heteroplasmy (frequency of deleterious mutant organismal DNAs, such as mitochondria) allows purifying selection to purge deleterious mutations, thus slowing Muller's ratchet. The most familiar mechanism employed by organisms to create such variance is a genetic bottleneck that accompanies the formation of the germ line. But many eukaryotes do not sequester a germ line early in development, which prevents the use of this mechanism. In this article, the authors investigate many proposed mechanisms that can increase cell-to-cell variance in heteroplasmy, and they incorporate all of them into a single, comprehensive genetic model that tracks the dynamics of heteroplasmy variance.

The model incorporates the following mechanisms: cell division, turnover, gene conversion, cellular subsampling, and reamplification. The authors use data from mouse models to show that their theoretical predictions match empirical measurements. Having confirmed the compatibility of the model with real-world measurements, the authors highlight some of the interesting points of their model's behavior, such as the opposite effects of organelle fission-fusion state. They point out that gene turnover increases variance when molecules are in fission state (because smaller fragments can undergo autophagy), whereas gene conversion increases variance when molecules are in fusion state (because fused molecules are available for recombination). Another model result is that for most mechanisms, variance increases more when there are few oDNA molecules (this is intuitive for the bottleneck case, since its effect results from sampling error); in contrast, gene conversion increases variance at a rate independent of copy number. From this, the authors make the prediction that in organisms that cannot resort to bottlenecks (i.e. those without early germ line sequestration) gene conversion can be important. They confirm this prediction by finding that, across the tree of life, organisms without a fixed body plan typically possess mtDNA recombination genes, which organisms with a fixed body plan lack. Further empirical evidence is provided by Arabidopsis, which has fragmented mitochondria (low conversion) -- except in the shoot apical meristem, which gives rise to the germ line, where mitochondria are fused. mtDNA recombination is also modulated in plants by surveillance genes; the authors find that, in agreement with this line of reasoning, these genes are highly expressed in the SAM.

I found this manuscript to be a highly impressive piece of work. The analytical work and predictions thereof are interesting enough on their own (for instance, the clarification of the different pathways by which the fission-fusion state of mitochondria may increase or reduce variance), but I'm particularly well-impressed with the extensive use of empirical data to validate the model (mouse models) as well as to test predictions made by the model (mtDNA recombination genes across the tree of life / the fission-fusion state of mitochondria in different tissues of Arabidopsis / the gene expression of surveillance factors in SAM in many species of plants). I don't often read articles that so thoroughly incorporate theory and data into a satisfactory and complete story.

I am not an empiricist, and hence cannot fully vouch for the quality or appropriateness of the empirical work. On those points, I defer to the opinion of other reviewers/editors who may be more familiarized with the relevant literature and methods. But with regard to the areas I feel competent to judge, I have no major requests for revisions, and recommend the article for publication. My only minor requests have to do with the clarity of the exposition.

I found parts of the introduction confusing: particularly, it was hard to identify precisely which mechanisms were going to be incorporated into the model and what those mechanisms consist of. The first such mechanisms are introduced in paragraph 4: it discusses recombination and fusion. Paragraph 5 again returns to recombination, and introduces the concept of gene conversion for the first time (but the relationship between these is only explained much later, in the Methods, in line 108). The other mechanisms are only introduced later: turnover only in line 135, reamplication in line 106. It is not until the third paragraph of the Results section that we get a straightforward list of mechanisms. I think that it would be extremely helpful to clearly define all mechanisms (partitioning at cell divisions, gene conversion, turnover, subsampling, and reamplification) early on in the Introduction, maybe in bullet form, with accompanying definitions. In doing so, please be mindful of readers who may not be entirely up-to-date with terms such as reamplification, conversion, DNA turnover, or the concept of mitochondrial fission/fusion.

I did not fully understand some parts of the procedure used when fitting the model to the mouse data. For instance, it was not clear to me why the authors present Fig. 1D (variance and bottleneck size for measured data and model prediction) for the mouse model HB but have no equivalent figure for mouse model LE. 

As far as I can understand, the authors fit the model (without selection) using mechanisms (i) cell division, (ii) turnover, and (v) reamplification, to mouse model HB. The text in Fig. 1D (and in lines 169 onward) shows how the model with all three mechanisms performs better than other, simpler models. But the text doesn't discuss a similar model comparison for the LE model. Instead, the text describes a model comparison between a model with and a model without selection. Are turnover, reamplification and cell division the best predictors for LE, as they were for HB? I understand that the authors wanted to test whether their model could correctly predict that selection played a role in the LE mice -- but would the model also predict selection in the HB mice? To be clear, I am not claiming that there is anything conceptually incorrect with the analysis, but I would prefer if the text were less synthetic and clearly justified the reasoning behind the different choices.

Smaller points:

The Discussion emphasizes the different mechanisms used by metazoans with early-sequestered germlines vs organisms without a fixed body plan, and the important role gene conversion plays for the latter. I think this is a fairly central take-home message, and should be included in the Author Summary (as it is in the Abstract).

Around line 44 - I think you should cite and discuss some of the classical models that introduce the idea that bottlenecks facilitate the purging of deleterious mutations -- at least Kondrashov (1994, Genetics) and Bergstrom & Pritchard (1998, Genetics)

Line 49 - Define Muller's ratchet and mutational meltdown

Ln 292 - typo, *in shoot*

Reviewer #3:

[identifies himself as Robert Lanfear]

This paper presents a new theory on the generation of variance in levels of heteroplasmy in organellar genomes. It is an amazing paper. I think 'tour de force' is an appropriate term here. The paper presents a new, and very clear, model. It then shows that the model fits observational data (where that data is available). It discusses a range of important implications of the model for evolutionary biology. It then performs two quite amazing empirical analyses (a comparative analysis across many taxa, and a gene expression analysis across a range of plants). I think most people would think of the work in this paper as being more suited to a series of at least three papers, but combining them together into a single concise and well-argued paper is (in my opinion) exactly the right thing to do. I enjoyed reading it enormously, for the rigour and scope of the science, the clarity of the writing, and the almost incredible concision. I have something to aspire to.

Having said all of that, I of course have a few comments that I hope might help to improve the paper a little. Many of them, I suspect, are due to my misunderstanding. Still, hopefully this can provide some direction in adding a few additional explanations here and there. 

Rob Lanfear

Minor

Line ~42: A completely niggly point, but I don't think the ongoing presence of mtDNA diseases proves that the shifts don't remove mutants: they could be due to recurrent denovo mutations (particularly given the high mutation rates of mtDNA, and the extremely variable mutation rates among sites). Perhaps change 'can't' from 'are rarely likely to' and link it to the evidence with something like "which perhaps helps to explain"?

It wasn't all that clear to me what the authors consider to be heteroplasmies? It seems like the assumption is that heteroplasmy refers to base-level differences between organellar genomes. But there are also structural heteroplasmies which can have similar effects. The generation of these two types of heteroplasmy can be very (very) different though.

 It would be useful to make explicit exactly what is being modelled. 

The list of processes in the paragraph that starts on line 104 seems to omit mutation between genotypes in the model. For single-base changes this seems reasonable (and anyway, you have to start somewhere) but some structural heteroplasmies are very stable because the mutation rates (if you can call them that, perhaps 'switching rates' is better) are very high. Clarity on what types of variation are in scope for the model would help here (see previous comment). 

Line 130: I was confused about how the influence of nonzero selective differences could be accounted for 'where neither allele experiences a selective advantage'. Perhaps the authors could clarify the conditions in the ms at this point, which I assume have to do with whether selection is operating inter- or intra-cellularly?

In the numbered list starting on line 133, 'n' denotes oDNA copy number in point (i), but 'population size' in point (ii). Can you clarify? It would be nice to keep the notation consistent, and I assume that it is, such that 'n' in point (ii) should be oDNA copy number. If 'n' in point (ii) isn't oDNA copy number, can you (a) use a different symbol, and (b) explain what the population size refers to (e.g. oDNA molecules, cells, clusters, etc).

On line 150 I couldn't figure out why 'genetic bottleneck' was in quotes. If you think it's not really a genetic bottleneck in the traditional sense (I can see arguments both ways) perhaps just call it something like the oDNA bottleneck? 

I would like to applaud the authors for Box 1. It's so clear! As someone who doesn't find large equations intuitive to understand, the annotations make all the difference.

Line 176: is it possible to compare (e.g. with AIC, R-squared) the NZB-BALB/c model to the current model? If not, it would be nice to explain why not. On a similar note, I would like to know the numbers behind the statement that the bottleneck size and estimate of the combined statistic vf/n 'agree well' with the analysis of the NZB-BALB/c model.

Line 181: what is the meaning of 'direct' in 'direct quantitative demonstration'. I agree that this is a quantitative demonstration, but I don't know the difference between a 'direct' and a (presumably) 'indirect' demonstration.

Line 222: I had to look at ref 49 to figure out what a 'kiss and run' event was. It's a cute name, but I think the paper would be clearer if it included a parenthetical explanation of what a kiss and run event is. I note also that it's hyphenated in ref 49 but not in this ms, and that ref 49 suggests (I didn't follow the next level of references) that the standard understanding of kiss and run events is from vesicles fusing with the cell membrane, rather than organelles (or bits of them) fusing with each other. More reasons for a parenthetical explanation I would argue.

Line 258: I love this hypothesis. It clearly flows from the model and is well explained. I can see how it could apply to e.g. mtDNA in plants (which is nicely explained in the next section), but I had trouble reconciling it with cpDNA in plants. I wonder if the authors could comment (here, or in the paper, preferably the latter but I know how it goes with every reviewer asking for extra things in the manuscript) on this? Specifically, given the limitations that plants are thought to have, how could they potentially solve the problem of cpDNA heteroplasmy variance generation?

I think the phylogeny in Figure 2 adds very little. It's very hard to see the relationships or the number of tips, and it's also very hard to figure out how the clades named in the tree match up with the gene tracks or the pictures below. I'd love to see something simpler here, perhaps just a tree that relates the major named clades, and some clearer delination between those clades that allows readers to see which parts of the gene tracks correspond to each clade.

The mtDNA work starting on line 261 is stunning, and frankly could be its own paper or series of papers. So please ignore this comment if it is inconvenient. As it is I think the conclusions being made (which are aknoweldged as being quite speculative) are perfectly appropriate from the analysis presented. Nevertheless, I would love to see something about the independence of the observations here. For example, a lot of the figure is devoted to showing that metazoa lack the first three genes listed, but that is really just a single evolutionary observation. There appears to be a lot more interesting stuff going on e.g. in Fungi, where sub-clades seem to be lacking all four genes. But even among the eight clades of interest I'd love to be able to better understand the extent to which the presence/absence of the genes of interest is correlated with development in an evolutionary sense. This could be done through a formal comparative analysis (overkill I think, given the scope and current length of the paper) or at least through a clearer tree in Figure 2 that would at least allow readers to get a better feel for how correlated these traits may. 

Line 292: "occurs shoot" should be "occurs in shoot"

The observations, analyses, and arguments in the section on recombination genes in plants were excellent. It strikes me that these observations also seem consistent with the 'functional germline' hypothesis, i.e. that the SAM operates in a way that makes it functionally rather similar to a segregated germline in animals. (There are lots of references about the functional germline hypothesis in the paper of mine that you already cite: ref 22). 

Major

In the paragraph starting on line 40 there seems to be some tension and a lack of clarity in between the general (e.g. talk of oDNA in general, and mtDNA and plastid DNA), and the specific (e.g. a lot of talk of mammals and mtDNA). I think some clarification is warranted. The previous paragraph talks very generally about the heteroplasmy level in oDNA. And this paragraph starts with a general statement about heteroplasmy level in the germline. But then almost all of the rest of the paragraph (and all but one or two references) seem to be about mammalian mtDNA. I think it would be very useful to be as explicit as possible about how major clades (e.g. plants, animals) and organelles (mitochondria in animals, mitochondria in plants, and plastids) relate to the seemingly general statements (or alternatively, and perhaps more likely, be explicit about which clades and organelles the conclusions are being drawn from, given that often the data will be pretty patchy). Of note, this issue seems to be limited to this paragraph. The rest of the paper is very explicit about which clades and organelles are the subject. 

I find the analyses in figure 1D convincing on visual inspection, but Figures 1C and 1E much less so. Perhaps it's just that there's a lot of overplotting, making it hard to guess the true variance from the figure (or maybe I haven't understood the transformed h properly)? With that caveat though, it does look visually like a constant variance model might fit better than the model from Eq 1 where the 95% intervals increase over time. With that in mind, I'd really like to see something more quantitative than the statement on line 171 that "the overall model captures well the time behaviour of observed single-cell heteroplasmy level distributions". Ideally this quantification would include some absolute measure of fit (e.g. of the amount of variation in the data captured by the model) as well as some comparison to sensible null (aka completely uninteresting models), and/or a purely descriptive model that shows the 95% confidence intervals in transformed h that you get directly from the data (it does seem like this might be exactly what's in Figure 1D, but I had a hard time figuring out if this is actually the case). In short, I remain a little bit less than convinced that the conclusion "the overall model captures well the time behaviour of observed single-cell heteroplasmy level distributions" is supported by the data presented. Perhaps all I need is a bit more hand-holding (e.g. if Figure 1D is already doing most of what I ask for). But even in that case I'd like to see an absolute measure of fit for 1D, like an R-squared or similar. AICs are all very well, but there will always be a model with a lowest AIC…

---

## [Editor Report · Decision Letter 2]

23 Feb 2021

Dear Dr Johnston,

On behalf of my colleagues and the Academic Editor, Tom Kirkwood, I'm pleased to say that we can in principle offer to publish your Research Article "Avoiding organelle mutational meltdown across eukaryotes with or without a germline bottleneck" in PLOS Biology, provided you address any remaining formatting and reporting issues. These will be detailed in an email that will follow this letter and that you will usually receive within 2-3 business days, during which time no action is required from you. Please note that we will not be able to formally accept your manuscript and schedule it for publication until you have made the required changes.

PRESS: We frequently collaborate with press offices. If your institution or institutions have a press office, please notify them about your upcoming paper at this point, to enable them to help maximise its impact. If the press office is planning to promote your findings, we would be grateful if they could coordinate with biologypress@plos.org. If you have not yet opted out of the early version process, we ask that you notify us immediately of any press plans so that we may do so on your behalf.

Thank you again for supporting Open Access publishing. We look forward to publishing your paper in PLOS Biology. 

Sincerely, 

Roli Roberts

Roland G Roberts, PhD 

Senior Editor 

PLOS Biology